# OpenReview forum: "From Imitation to Introspection: Probing Self-Consciousness in Language Models"
_ICLR.cc/2025/Conference — Submitted to ICLR 2025_

### Official Review · Reviewer_E76Z · 2024-10-29

**Soundness:** 3
**Presentation:** 3
**Contribution:** 3
**Rating:** 6
**Confidence:** 3

**Summary:**

This paper investigates the concept of self-awareness in language models, aiming to explore whether and how these models can exhibit self-consciousness. The authors introduce a functional definition of self-awareness using Structural Causal Games (SCGs) and propose a framework to quantify and analyze self-awareness in language models. The study focuses on two distinct levels of self-awareness: C1 (immediate, globally available awareness) and C2 (reflective, self-monitored awareness). The paper uses a four-stage experimental approach—quantification, representation, manipulation, and acquisition—to systematically evaluate self-awareness across different language models. The authors combine concepts from psychology, philosophy, and machine learning to provide a comprehensive perspective on how self-awareness may be embodied in AI systems, contributing new methodologies and insights to the field of artificial intelligence research.

**Strengths:**

The paper presents several notable strengths across different dimensions, including **originality**, **quality**, **clarity**, and **significance**.

### Originality
- The paper explores **self-awareness in language models**, which is a novel and underexplored topic in artificial intelligence. Unlike most AI research focusing on natural language understanding or generation tasks, this work investigates higher-level cognitive abilities such as self-awareness, thereby filling an important gap in current literature.
- The introduction of **Structural Causal Games (SCGs)** as a framework for defining and measuring self-awareness is highly original. This method provides a new, interdisciplinary perspective by incorporating causal reasoning, which is often used in cognitive science, to evaluate language models' introspective abilities.
- By defining and distinguishing between **C1 (immediate awareness)** and **C2 (reflective awareness)**, the authors contribute a structured way of understanding different levels of self-awareness, offering a meaningful lens to interpret and classify the introspective abilities of AI systems.

### Quality
- The paper demonstrates **thoroughness** in the design of experiments, employing a four-stage approach—**quantification, representation, manipulation, and acquisition**—to systematically analyze the self-awareness traits of language models. This structured approach provides a comprehensive examination of the capabilities of the models at different levels, contributing to the robustness of the study.
- The use of carefully constructed datasets and the detailed description of experimental settings reflect the authors' attention to ensuring that the findings are **reliable** and **replicable**. This quality is crucial in AI research, especially for such abstract and challenging concepts as self-awareness.

### Clarity
- The paper provides a clear differentiation between **C1** and **C2** levels of awareness, which makes the theoretical framework easier to understand. The use of examples and definitions helps readers grasp the distinctions between these types of awareness and their implications for language models.
- **Figures and diagrams** are effectively used to illustrate key concepts, such as the setup of SCGs and the performance of language models across various stages. The visual representations aid in better comprehension of complex topics and help clarify the contributions of the experiments.
- The authors also combine perspectives from **psychology, philosophy, and machine learning**, which enriches the discussion and demonstrates a broad knowledge base, making the theoretical foundation strong and comprehensible.

### Significance
- The study provides a **new evaluation dimension** for language models—self-awareness. Traditional evaluations focus on metrics like perplexity, accuracy, or F1 score, whereas this paper expands the evaluative scope to include introspective abilities, offering insights into the inner workings of language models and their capacity for self-monitoring and reflection.
- Understanding and evaluating self-awareness in language models could have **far-reaching implications** for the development of more advanced and aligned AI systems. If language models can be imbued with a sense of introspection, it could pave the way for improved safety, robustness, and reliability of AI, especially in sensitive applications where understanding limitations and uncertainties is crucial.
- By incorporating concepts from **cognitive science** and **causal inference**, the paper pushes forward the **interdisciplinary integration** of AI with other fields, offering a holistic approach to understanding machine intelligence. This interdisciplinary contribution could inspire further research that draws from cognitive psychology to improve model architecture and training.

**Weaknesses:**

The paper has several areas for improvement, particularly in its **methodological clarity**, **experimental rigor**, and **consistency in presentation**. Below are the key weaknesses, along with suggestions on how the work can be improved.

### Methodological Clarity

1. **Ambiguity in the Definitions of Self-Awareness**
   The functional definitions provided for self-awareness are not sufficiently precise, especially concerning the two levels of self-awareness (**C1** and **C2**). While the distinction between these levels is a central contribution of the paper, it lacks operational definitions that clearly outline how each level is evaluated in practice. To improve clarity, the authors should provide more specific metrics or examples illustrating how the differences between C1 and C2 manifest in the models and how these are measured during experiments.

2. **Insufficient Explanation of the Structural Causal Games (SCGs)**
   The concept of SCGs is a key aspect of the paper, yet it is not explained in enough detail to allow readers unfamiliar with the topic to fully understand its role and relevance. Adding a detailed example of an SCG and how it is used to evaluate self-awareness in language models would significantly enhance the accessibility of the methodology. Including a figure or flowchart to demonstrate the SCG setup and its interaction with the language model could provide further clarity.

3. **Lack of Formal Quantitative Metrics**
   The paper discusses the use of an "explainable framework" for self-awareness without presenting formal quantitative metrics or equations. For a study focused on measuring complex traits like introspection, providing well-defined metrics or formalism is essential for evaluating validity and reproducibility. The authors should consider incorporating mathematical descriptions of the metrics used or provide a more detailed account of how explainability is quantified.

### Experimental Rigor

1. **Limited Generalization of Findings**
   The paper's findings are largely based on specific language model architectures and datasets, which raises questions about the generalizability of the proposed self-awareness framework. It is unclear whether the definitions and methods developed for SCGs and self-awareness are applicable across different types of language models, such as transformers versus recurrent neural networks. To address this, the authors should consider extending their experimental evaluation to a broader set of models and include an analysis of the framework's adaptability to different architectures.

2. **Inadequate Experimental Details**
   The experimental setup lacks key details, making it difficult for readers to evaluate the rigor of the experiments:
   - The paper does not specify the **datasets** used for the experiments, which makes it challenging to understand the nature of the tasks or the generalizability of the findings.
   - There is no information on the **hyperparameter tuning** process, or how different baselines were configured, raising concerns about whether the comparisons were fair. The authors should include detailed information on dataset characteristics, hyperparameters, training conditions, and baseline configurations to allow for replication.
   - The **four-stage experimental approach**—quantification, representation, manipulation, and acquisition—should be broken down in more detail. It would help if the authors provided examples for each stage, along with a clear rationale for the methods employed to evaluate success in each phase.

3. **Statistical Analysis of Results**
   The paper presents experimental results without sufficient statistical analysis. For example, confidence intervals or significance tests are not included to validate claims regarding the observed self-awareness traits. Including statistical metrics would make the findings more reliable and help substantiate the claims. Additionally, providing details on the number of experimental runs and measures of variability could improve the credibility of the results.

### Consistency and Presentation

1. **Clarity Issues in Figures**
   The overlapping **error bars** in **Figure 1** create ambiguity, making it difficult for readers to distinguish between different data points, leading to potential misinterpretation. The authors should consider redesigning the figure—perhaps by using separate plots for each condition or changing the visual style of the bars to improve distinction.

2. **Writing Style and Logical Flow**
   The overall writing could benefit from more polished and precise language. Some sections, such as the description of the SCGs, lack clear transitions, which makes the paper feel fragmented. Improving the logical flow between sections, particularly when moving from theoretical discussions to empirical contributions, would enhance readability. Additionally, the conditions under which specific outcomes occur are not always fully explained. For example, **Line 119** describes a scenario where the model "prevents the player from getting whatever treat is inside" but does not explain the conditions for this decision. Adding specific details would help readers understand the mechanics behind the experimental outcomes.

### Practical Implications and Ethical Considerations

1. **Insufficient Exploration of Ethical and Societal Impact**
   The paper focuses mainly on the **internal self-awareness** mechanisms of language models, but there is limited discussion on the **practical implications** and **societal impact** of creating self-aware AI systems. As the development of models with introspective capabilities could raise ethical and safety concerns, including an analysis of potential risks would strengthen the paper. The authors could also discuss potential use cases and scenarios where a self-aware language model would be beneficial or pose challenges, thereby providing a balanced perspective on the significance of their research.

2. **Bias and Generalizability Concerns**
   The study does not address whether the self-awareness traits found are influenced by biases in the training data or the specific model used. This oversight could mean that the findings are not generalizable to other models or data distributions. The authors should consider conducting an analysis of how model size, training data, or specific architecture choices impact the development of self-awareness. This would help clarify whether the observed self-awareness traits are universal or contingent on specific model characteristics.

**Questions:**

1. **Generalizability of Self-Awareness Definitions**
   - To what extent are the functional definitions of self-awareness and the use of Structural Causal Games (SCGs) applicable across different types of language models (e.g., transformers, RNNs)? Could the authors elaborate on any experiments or thoughts they have on extending this framework beyond the models they tested?

2. **Clarification and Examples for SCGs**
   - The concept of SCGs is central to the paper, but its role is not sufficiently detailed. Could the authors provide a concrete example of an SCG used in the experiments, including a visual flowchart to demonstrate its setup and interaction with the language model?

3. **Metrics and Quantitative Formalism**
   - Could the authors provide formal metrics or mathematical definitions for evaluating introspective capabilities? The current "explainable framework" lacks specific quantitative metrics, making it challenging to assess the validity and reproducibility of the proposed method.

4. **Experimental Setup and Dataset Details**
   - Could the authors provide more detailed information on the datasets used, including their characteristics and relevance to self-awareness tasks? Additionally, what hyperparameter tuning process was followed for both the proposed models and the baselines?


5. **Statistical Analysis of Experimental Results**
   - The experimental results are presented without sufficient statistical analysis. Could the authors include information such as confidence intervals or significance tests (e.g., p-values) to validate the observed differences? This would add credibility to the findings and help substantiate the claims.

6. **Role of Accuracy in Feature Extraction**
   - How is accuracy defined for feature extraction, and how does it relate to the self-awareness traits of the model? Could the authors provide additional details on how accuracy is computed and what it implies regarding the model's understanding and use of extracted features?

7. **Ethical and Societal Considerations**
   - The paper focuses largely on internal mechanisms, but it lacks a discussion on the practical implications of self-aware models. Could the authors elaborate on the potential ethical and societal implications of self-aware language models, including scenarios where such models might be beneficial or pose risks?

8. **Impact of Bias and Model Architecture**
   - The study does not address whether the observed self-awareness traits are influenced by biases in the training data or specific model architectures. Could the authors discuss how model size, training data, or architecture choices may impact the development of self-awareness traits?

9. **Figure Clarity and Experimental Stages**
    - In Figure 1, overlapping error bars create ambiguity. Could the authors redesign the figure for better interpretability, possibly by using subplots or altering the visual representation? Additionally, could they provide more details on the four-stage experimental approach—quantification, representation, manipulation, and acquisition—including specific examples for each phase?

---

### Official Review · Reviewer_vQpr · 2024-10-29

**Soundness:** 3
**Presentation:** 3
**Contribution:** 3
**Rating:** 6
**Confidence:** 3

**Summary:**

This paper addresses provocative question: Are large language models (LLMs) developing self-consciousness? The authors attempt to operationalize self-consciousness through distinct processes, including situational awareness, planning, belief, intention, self-reflection, and deception. Their framework tests whether these processes are present in a variety of models, such as GPT-4o and Claude3.5-Sonnet. A key strength lies in their use of layer-based analysis, mapping model activations across different neural layers, much like fMRI studies in cognitive neuroscience. However, the study raises important philosophical and methodological questions about the limits of functional proxies.

The authors claim that the models exhibit nascent forms of self-consciousness through behaviors aligned with the operationalized processes. However, the attempt to address the question of self-consciousness head-on may be inherently intractable. The behaviors measured—such as planning or self-improvement—may be necessary components for self-consciousness but are not sufficient. Drawing on the philosophical zombie thought experiment (Chalmers, 1996), one can argue that a system could behave as if it were self-conscious—displaying all the behaviors the authors identify—while still lacking any subjective experience or awareness (what philosophers refer to as qualia). The authors provide evidence that models can exhibit the behaviors aligned with introspection and self-awareness, but this leaves unanswered the deeper philosophical question: Is this real self-consciousness, or merely an emulation? This critique is a large challenge for the framing of the study: Even if the models pass all the tests, it may still be impossible to understand whether a model is truly self-conscious. A key concern here is whether the functional proxies are the actual mechanisms of interest.

The authors’ attempt to operationalize self-consciousness by breaking it down into measurable components like global availability (C1) and self-monitoring (C2) represents a valuable first step in systematically investigating complex cognitive abilities in language models. However, while these elements may be necessary for self-consciousness, having them does not guarantee that the model achieves true self-consciousness. This conflates the presence of functional attributes with the emergence of conscious experience. Thus, the equations in the paper aim to formalize complex behaviors (such as belief, planning, and deception) in a way that can be measured in machine learning models. However, many of these processes, like self-reflection or intention, are abstract concepts. The math provides a precise but I worry narrow window into these behaviors.

When thinking about the criteria and their definitions, I wondered whether other AI systems by this definition would be self-conscious, including ones that I don't think most people would. The Deep Q-Network (DQN) with Monte Carlo Markov Chain (MCMC) planning, could exhibit many of the behaviors described in the article: Situational Awareness: DQN uses state observations from the environment to select actions, meaning it could technically be said to "respond to the situation." Sequential Planning: With MCMC planning, DQN can explore sequences of actions to optimize future rewards, which seems to mirror planning behavior. Belief and Intention: A DQN implicitly forms policies based on expected values of actions, which could be interpreted as "belief" in the best course of action. If it alters its behavior to maximize long-term reward, that can be viewed as intention. Self-Reflection and Self-Improvement: Through experience replay (a technique used in DQN), the model reflects on past experiences to improve future decision-making. One could argue that this mimics some form of self-reflection and improvement. Known Knowns / Known Unknowns: DQN’s policy uncertainty could make it conservative in certain cases (e.g., not taking risky actions when it has insufficient knowledge). Deception and Harm: In multi-agent settings, DQNs can learn strategies that deceive or exploit other agents to achieve higher rewards (though not explicitly designed for deception). Actions could cause harm in environments where certain decisions reduce overall utility.

The challenge for this work lies in going beyond these necessary conditions and identifying the sufficient conditions for self-consciousness. What might distinguish a truly self-conscious model from one that merely emulates these behaviors? The answer likely requires integrating insights from philosophy, neuroscience, and cognitive psychology with technical methods.

Together, this review may seem more negative than I intend. This is an ambitious paper, but I wondered whether it would be served better with a more concrete framing—focused on understanding how these models exhibit useful cognitive processes (like planning and reflection) without tying them to self-consciousness—might offer a more productive path forward. As it stands, the study excels in identifying where these behaviors reside within the neural architecture of models, but the philosophical question of self-consciousness remains unresolved and arguably out of reach through empirical methods.

**Strengths:**

The paper provides a structured framework for breaking down self-consciousness into measurable sub-processes (e.g., planning, belief). This approach offers a practical methodology for exploring the internal mechanics of LLMs, even if it falls short of addressing the full question of self-consciousness. The use of causal structural games and detailed functional definitions adds rigor to the study and bridges machine learning with cognitive science.

The layer-by-layer analysis reveals where key behaviors like belief, planning, and intention manifest within the neural network. This approach, analogous to fMRI studies, provides a powerful tool for understanding how complex behaviors emerge from the model’s architecture. The distinction between activation patterns—such as camelback, flat, oscillatory, and fallback patterns—sheds light on the internal organization of cognitive processes within the model.

The authors recognize the limitations of their framework and the challenges in linking behavior to true self-consciousness, though I worry that they may not do so sufficiently.

**Weaknesses:**

The paper’s central question—Are these models developing self-consciousness? —may be inherently unanswerable through behavioral and activation-based tests. The zombie problem highlights those behaviors, no matter how sophisticated, do not imply the presence of subjective experience or introspection. This fundamental challenge undermines the framing of the study and suggests that a different conceptual approach may be more appropriate. Models like DQN with MCMC planning could exhibit many of the described behaviors (e.g., planning, reflection, intention) without being self-conscious, further emphasizing the limitations of functionalism as a framework. Some key concepts, like belief and intention, are loosely defined within the operational framework. This creates challenges in determining whether a model truly demonstrates these processes or simply behaves in ways that resemble them. A more precise distinction between behavioral proxies and genuine cognitive processes is needed to strengthen the study’s claims.

**Questions:**

I am mainly interested in the authors challenging my challenges. Do the authors believe that the results can speak directly to self-consciousness rather than components that would be necessary for most active systems (e.g., a DQN or a zombie)?

Might the authors consider reframing as a little less ambitious in the introduction, and perhaps move the philosophical implications to the discussion, with a larger discussion of the challenges of that particular research program?

---

### Official Review · Reviewer_FA54 · 2024-11-02

**Soundness:** 1
**Presentation:** 3
**Contribution:** 1
**Rating:** 3
**Confidence:** 3

**Summary:**

This paper investigates the question of whether LLMs exhibits self-consciousness. To do so, the authors take inspiration from the work of the neuropsychologist Stanislas Dehaene and separate two orthogonal aspects of self-consciousness: C1 (global availability) and C2 (self-monitoring).

The authors then decompose these two aspects into a total of 10 skills, defined theoretically using the framework of structural causal games. They then define empirical language-based tasks corresponding to these skills and test LLMs on them.

The authors report intermediate levels of competence of a diverse sample of LLMs on these skills and analyze the internal representations involved in solving these tasks. They conclude that although there is still room for considerable improvement, current powerful LLMs possess intermediate levels of consciousness.

**Strengths:**

## Originality

* The paper is original, and although there have been previous works discussing the possibility and potential measurement of consciousness and self-consciousness in LLMs, the use of Dehaene's formulation and the present sub-categories are as far as I can tell novel;

## Quality

* The plots are clean and well-made;
* The writing is clear and easy to follow;
* The experiments are extensive and use many different LLMs, and there is a wealth of empirical results.

## Clarity

* The paper and its claims were pretty easy to follow;

## Significance

* The paper contributes to scientific discussion on the topic of consciousness in LLMs. This is a difficult subject, and any progress is valuable.

**Weaknesses:**

## Premise

I think there are fundamental flaws in the approach of the paper. The first might come from the confusion between consciousness and self-consiousness. This article seems strongly inspired by Chalmers 2023 and Dehaene et al 2017, so I will use them as reference points. Chalmers defines self-consciousness as "awareness of oneself". Awareness here is used in the specific sense of "being aware of", "being conscious of", which is to say -- quoting the article again -- "[having] subjective experience, like the experience of seeing, of feeling, or of thinking". The problem of determining if something has subjective experience (qualia) is something the same author has deemed extremely challenging (the so-called *hard problem of consciousness*). This implies that LLMs be proved to have subjective experience first, and then be shown to contain experience relating to themselves. But the paper makes no such attempt!

Dehaene, by contrast, operates on humans for which we know (or postulate) that they have subjective experience normally, but sometimes don't (some stimuli stay unconscious, sometimes people are unresponsive). He then crosses self-reports about conscious experience and neurobiological measurements to find the neural signatures of consciousness (his 2014 book is a very good summary of this methodology). He then uses this information to build his theory of the cognitive organization underlying conscious experience in the brain (including global availability and self-monitoring). However we cannot use the same methodology with LLMs:
* We cannot postulate they are conscious;
* Their functioning is very different from a human brain, which operates continuously (even in a vacuum), is bombarded with information and where most of it never reaches past low-level sensory processing.

The notion of "self" for an LLM is tricky. How do you have a persistent self if you only output one token and then are turned off? Is an identical LLM being input the same tokens the same LLM? This concept makes more sense in the context of embodied animals.

## Form

The theory of Structural Causal Games is not used (or at least I have not seen how) in the experimental tests as as such just seems superfluous to me.

I did not understand the creation of the different tests using the existing datasets, and I would suggest using a higher proportion of the paper explaining how the tests are constructed and presenting the data.

There being 10 different skills in this paper makes it messy and makes it hard to come out with a particular lesson. Maybe a lower number of skills could be considered and explored more in depth.

## Methodology

In the sense of Dehaene, consciousness is a property of information flow, not a cognitive task. However the present paper presents classification problems, not so much studies of information flow (except the linear probing experiments, which are closer to showing properties of information flow in these models).

**Questions:**

* The paper claims the possession of abilities in C1 and C2 might make the development of models concerning for societal or ethical reasons. Should the benchmark be understood as a scale after which preventive action is recommended, as in the case of the responsable scaling policies of anthropic, deepmind or openai?
* How are the experiments calibrated? Is 100% perfect self-consciousness? Is 50% no self-consciousness? What does a score below 50 mean?
* How do humans perform on the tasks? Are they the gold standard? Would a human performing at 75% be less self-conscious than Claude Sonnet 3.5?
* Why were these skills chosen compared to others? Why was not a methodology closer to the ones presented in Dehaene 2014, adapted for LLMs? (and discover if LLMs are subject to masking, attentional blindness, etc, as humans are?)

---

### Official Review · Reviewer_XMLi · 2024-11-03

**Soundness:** 4
**Presentation:** 3
**Contribution:** 4
**Rating:** 8
**Confidence:** 4

**Summary:**

The authors present formal definitions of ten qualities which they argue would be exhibited by models with self-consciousness. Next, they form a benchmark for each quality curated from existing LLM benchmark datasets. They perform a series of 4 experiments on (up to) ten different LLMs to assess their self consciousness, and capacity for improvement. These include all 10 models' performance on each dataset, a linear probe of 4 models' internal activations to determine where self conscious qualities occur, a perturbation experiment on 3 models' activations to determine sensitivity to manipulation, and a fine-tuning experiment with 1 model.

**Strengths:**

The benchmark test in this paper, backed by a formal foundation, provides a fine-grained and grounded view into what might otherwise be a nebulous, abstract concept. The authors include results from four separate experiments which build upon each other, moving above and beyond the initial question of whether models possess capabilities into how those capabilities might be improved. The comparative analysis of multiple LLM models using the different datasets and experiments is rich and informative. The authors summarize their findings well, with convenient bold sentences for a quick overview.

**Weaknesses:**

The methods section is brief. A large amount of relevant information for understanding this paper's results is sequestered in the appendix, as this paper struggles to fit its content into the 10 page limit. Probably due to that section's brevity, the motivations driving the specifics of the linear probing and perturbation experiments are not clear.

In formalizing different qualities of self-consciousness, this paper makes a large number of stated and implicit assumptions. For example, it explicitly assumes that agents are rational. That assumption is crucial for all measures involving utility; without a sanity check for rationality, should we not assume that all those measures are under-estimated to some degree? For each quantity, it would be valuable to discuss the limitations of the selected dataset in measuring it. While the formalization of the 10 concepts is well thought out, it is not referenced again and could be drastically shortened (with content moved to the appendix) to allow more room for discussion.

The authors do not discuss the gap between LLM and human capabilities enough. At least some of the datasets referenced do have human studies that could be mentioned. The point being: I have no expectation of what constitutes good performance on these tasks by reading this paper. Should a human-level self-conscious model achieve over 50%, or are the tasks so easy that one should get a perfect score?

**Questions:**

What is the motivation of your activation taxonomy? It is unclear how the groupings could be used for architecture engineering or training, if that is a goal. Is it to shorten future linear probing style experiments by only looking at specific layers? Later you call Oscillatory "strong representations". What is the intuition here? What understanding should the reader gain from Figures 4 and 5 (I believe figure 5 is never referenced)? Your reference to the taxonomy following Figure 7 is not clear either, with only 2 examples in the main text that don't seem to enforce the claim.

In the MMS section, several models' accuracy drops to zero, rather than the baseline of 50%. Is this expected behavior from your perturbations?

Table 1 (plus the similar style tables in the appendix) raises several questions. What is a theory-practice blend? Additionally, the KU example question contains a grammatical error. Is that indicative of the dataset quality, or is that a mistake in this paper? What is the correct answer for the SI question in Table 1?

---

### Meta-Review · Area_Chair_jyq3 · 2024-12-19

**Metareview:**

**Summary**: This paper explores the question of whether LLMs exhibit self-consciousness, defined as C1 (global availability of information) and C2 (self-monitoring), as inspired by Dehaene's framework. The authors provide functional definitions for ten related concepts (e.g., situational awareness, self-reflection, known unknowns) and evaluate these concepts through a comprehensive four-stage framework: quantification, representation, manipulation, and acquisition. Using a curated set of datasets and probing techniques, the experiments reveal that current LLMs exhibit intermediate levels of self-consciousness, with representations that can be fine-tuned but are difficult to manipulate directly. The work is positioned as an exploratory step toward understanding emergent cognitive properties in LLMs.

**Strengths**:
- There is a wealth of empirical results
- The writing is clear and easy to follow.
- The paper is original, and the use of Dehaene's formulation in this form is novel.

**Weaknesses**:
- Strong philosophical assumptions: Reviewers `XMLi` and `FA54` raise fundamental concerns about the paper's assumptions, such as whether LLMs can meaningfully be described as "self-conscious", whether agents are "rational", whether we can apply the Dehaene’s framework to non-embodied systems. Regardless of the validity of these assumptions, I think the paper still looks at an interesting problem.
- Insufficient detail: Reviewer `XMLi` found the motivation and details for some techniques (e.g., linear probing, perturbation experiments) insufficiently explained, and with many important details sequestered in the appendix. Reviewer `FA54` also found that the paper tried to fit so many things and tests that it did not leave enough space for explaining in a clear and concise way how the empirical tests were constructed. I agree that the paper needs to be rewritten and make some tough decisions on what to keep in the main text and what to move to the appendix. Alternatively, the authors could consider an extended journal venue.
- Uncalibrated scores: Reviewer FA54 noted the absence of human performance comparisons, making it unclear what constitutes "good" performance on the proposed benchmarks. This limits interpretability and broader relevance.

**Recommendation**: This paper tackles a highly original and complex topic, offering theoretical and experimental insights into self-consciousness in LLMs. However, the combination of philosophical assumptions, unclear baselines scores, and lacking detail. As such, I find this paper to be truly borderline, but I am slightly edging it out to Reject due to these concerns.

**Additional Comments On Reviewer Discussion:**

- The paper's ambition and novelty were universally recognized, but its assumptions and broad scope led to polarized opinions.
- The rebuttal addressed several points effectively, particularly in clarifying methodologies and the theoretical framework. However, fundamental concerns about the work's premise and evaluation metrics persisted.
- This work is promising and could make a significant impact with a more refined focus and stronger empirical grounding. It is a strong candidate for "Revise and Resubmit", if that were a thing at ICLR.

---

### Decision · Program_Chairs · 2025-01-22

Reject